# An IgY Effectively Prevents Goslings from Virulent GAstV Infection

**DOI:** 10.3390/vaccines10122090

**Published:** 2022-12-07

**Authors:** Mengran Zhang, Lijiao Zhang, Jing Yang, Dongmin Zhao, Kaikai Han, Xinmei Huang, Qingtao Liu, Yichen Xiao, Youfang Gu, Yin Li

**Affiliations:** 1College of Animal Science, Anhui Science and Technology University, Fengyang 233100, China; 2Institute of Veterinary Medicine, Jiangsu Academy of Agricultural Science, Nanjing 210014, China; 3Key Laboratory of Veterinary Biological Engineering and Technology, Ministry of Agriculture, Nanjing 210014, China

**Keywords:** goose astrovirus, egg yolk antibody, prevention, treatment, passive immunoprevention

## Abstract

Goose astrovirus (GAstV) leads to viscera and joints urate deposition in 1- to 20-day-old goslings, with a mortality rate of up to 50%, posing a severe threat to entire colonies; however, there is no efficient prevention and control method for GAstV infection. This study describes a prophylactic anti-GAstV strategy based on the specific immunoglobulin Y (IgY) from egg yolk. The specific IgY was produced by 22-week-old laying hens intramuscularly immunized with the inactivated GAstV three consecutive times, with 2-week intervals. The egg yolk was collected weekly after the immunization and the anti-GAstV IgY titer was monitored using an agar gel immune diffusion assay (AGID). The results revealed that the AGID titer began to increase on day 7, reached a peak on day 49, and remained at a high level until day 77 after the first immunization. The specific IgY was prepared from the combinations of egg yolk from day 49 to day 77 through PEG-6000 precipitation. Animal experiments were conducted to evaluate the effects of prevention and treatment. The result of the minimum prophylactic dose of the IgY showed that the protection rate was 90.9% when 2.5 mg was administrated. Results of the prevention and the treatment experiments showed prevention and cure rates of over 80% when yolk antibody was administered in the early stages of the GAstV infection. These results suggested that the specific IgY obtained from immunized hens with the inactivated GAstV could be a novel strategy for preventing and treating GAstV infection.

## 1. Introduction 

Astroviruses (AstVs) are nonenveloped, single-stranded, positive-sense RNA viruses belonging to the *Astroviridae* family, divided into two genera as the susceptible host, namely, *Mamastroviruses* (MAstVs) and *Avastroviruses* (AAstVs) [1]. In 2015, a goose-derived astrovirus was discovered with the primary symptoms of visceral and joint gout in China [2]. Based on the homology analysis, the goose astrovirus (GAstVs) was divided into two genotype groups, goose astrovirus type 1 (GAstV-1) and goose astrovirus type 2 (GAstV-2) [3]. GAstVs mainly occur in 1- to 20-day-old goslings with the rate of infection and mortality of about 80% and 50%, respectively [4]. Recent studies have reported that GAstV could infect ducklings and chickens and cause the typical symptoms of gout disease [5,6,7]. The widespread of GAstV infection has leaded serious economic loss in the goose industry in China.

Egg yolk immunoglobulin (IgY) is the major antibody in avian. The molecule of IgY has a structure similar to that of IgG, with two heavy chains with a molecular weight of 67 to 70 kDa and two light chains with 25 kDa. IgY has four constant regions at heavy chains, while IgG has three constant regions [8]. Since the 1980s, IgY has been widely used for disease prevention and treatment owing to its safety, effectiveness, and stability [9,10]. Egg-derived IgY antibodies can be produced on a large scale and purified from egg yolk one or two weeks after the last immunization. One immunized hen can produce approximately 35 g of IgY yearly, ~100 mg of which is antigen-specific [11]. Additionally, numerous studies have identified the protective effect of IgY for viral infections in poultry, such as avian influenza virus [12,13], Newcastle disease [14,15], duck viral hepatitis [16], chicken infectious bursal disease [17,18], reovirus [19,20], and duck adenovirus [21]. There are no approved vaccines against GAstV infection so far, IgY could be a promising prophylactic treatment strategy. 

In this study, the anti-GAstV IgY was prepared using the laying hens immunized with the inactivated GAstV-AHQJ18 three times and used in clinical experiments. Results showed a good prophylactic and therapeutic effect against GAstV infection, which would provide a new strategy for preventing and treating GAstV infection.

## 2. Materials and Methods

### 2.1. Cell Culture and Virus

Leghorn male hepatoma (LMH) cells (Sunncell, Wuhan, China) were cultured in Dulbecco’s Modified Eagle’s medium (DMEM, Thermo Fisher Scientific, Shanghai, China) supplemented with 10% fetal bovine serum (FBS, Cytiva, Shanghai, China) and 100 U/mL streptomycin-penicillin (Thermo Fisher Scientific, Shanghai, China) in a humidified incubator with 5% CO_2_ at 37 °C. The GAstV AHQJ18 strain (Genbank accession: OP556137) belonging to GAstV-2 was propagated in LMH cells and the titer was determined as 10^5.84^ TCID_50_/0.1 mL and stored in our laboratory. 

### 2.2. Establishment of Challenge Model

Twenty healthy 1-day-old goslings were divided into two groups. Ten goslings were subcutaneously infected with 10^5.84^ TCID_50_ of the GAstV-AHQJ18 strain, as the GAstV-infection group; the remaining ten goslings were injected with sterile phosphate-buffered saline (PBS, pH 7.4) in the same manner and kept separated from the infected group, as the control group. The death of goslings was monitored on daily basis. Cloacal swabs were collected from five goslings in each group at 4-, 8-, and 12-days post-infection (dpi). The viral loads of swabs were detected by an SYBR Green Ⅰ real-time PCR (qRT-PCR) conducted by Luo [22]. The dead goslings were autopsied and tissue samples (liver, spleen, and kidney) were collected for histopathological analysis. The samples were fixed in 4% paraformaldehyde and sent to the Servicebio Company (Wuhan, China) for pathological sectioning, hematoxylin, and eosin (H&E) staining, and light microscopy examination.

### 2.3. Preparation of Inactivated Virus

The virus suspension was filtered through a 0.22-μm-pore-size filter (Millipore, Boston, MA, USA) to remove bacteria before formaldehyde inactivation. The modified inactivation was carried out by adding 37% formaldehyde (Sinopharm chemical reagent Co., Ltd., Beijing, China) to the virus suspension to give a final formaldehyde dilution of 1/1000 and incubating the suspension for 24 h at 37 °C [23]. The inactivated virus suspension was filtered as described above and stored at 4 °C. No live viruses were detected after repeated LMH cell cultures for five passages. Tween-80 (Sinopharm chemical reagent Co., Ltd., Beijing, China) was added at a 4% volume of the inactivated virus preparation, as the water phase. The mixture of the 94:6 volume ratio between No.10 white-oil (Sinopharm chemical reagent Co., Ltd., Beijing, China) and Span-80 (Sinopharm chemical reagent Co., Ltd., Beijing, China) was used as the oil phase, after sterilization at 121.3 °C. The volume ratio of the oil phase to the water phase is 2:1, and the mixture was homogenized by a homogenizer. The inactivated virus was produced as the W/O type emulsion.

### 2.4. Immunization of Laying Hens

Twenty-two-week-old white leghorn laying hens (*n* = 20 per group) (Jiangsu Lihua Animal Husbandry Co., Ltd., Changzhou, China) were immunized intramuscularly at three different sites with the inactivated virus or an aliquot of PBS; the immunization was performed three consecutive times, with 1 mL, 1.5 mL, and 2 mL respectively, and two-week intervals [24,25,26]. Eggs were collected from immunized hens each week for immunization evaluation throughout 21 weeks post-vaccination (WPV).

### 2.5. Assessment of Specific IgY Titers

An AGID assay was performed to determine the egg yolk antibody titer using the method described by Serena and Smith [27,28]. In brief, 70 μL 100-fold concentrated GAstV (7 × 10^6.84^ TCID_50_) was deposited in the center well, the egg yolk consisting of antibodies diluted 1:2, 1:4, 1:8, 1:16, 1:32, and 1:64 was added to the wells surrounding the center well. The lines of precipitin are formed at the sites of a combination of antigen and antibody after an incubation period of 12–48 h in a moist chamber.

### 2.6. Preparation of IgY Antibody

IgY was purified from the egg yolk mixture when the yolk antibody titer turned relatively steady. The IgY purification protocols, preparing a water-soluble fraction (WSF) of yolk and precipitating IgY in polyethylene glycol (PEG), were proposed by Akita and Polson [29,30]. Briefly, the egg yolk was solubilized in PBS. The yolk granules were pelleted by centrifugation at 10,000× *g* for 20 min at 4 °C, the supernatant (WSF) was filtered and added a concentration of 3.5% polyethylene glycol 6000 (PEG 6000, Biosharp, Beijing, China). After being stirred for 30 min at room temperature, the mixture was centrifuged at 10,000× *g* for 20 min at 4 °C. The supernatant (filtered through three layers of filter paper) was collected and added PEG 6000 to the final polymer concentration of 12% (*w*/*v*). The mixture was centrifuged at 10,000× *g* for 30 min at 4 °C and collected the pellets, resuspended in PBS subsequently. The purity and yield of the IgY were monitored by 10% sodium dodecyl sulfate-polyacrylamide gel electrophoresis (SDS-PAGE). The concentration of the IgY was determined using the Bradford protein concentration Assay kit (Beyotime, Shanghai, China).

### 2.7. Virus Neutralization Assay (VN)

The virus neutralization assay was conducted as previously described [31,32]. The egg yolk, collected from immunized hens each week, was titrated with thirteen dilutions (1:2^1^~1:2^13^) in DMEM in 96-well microtiter plates (Corning, NY, USA), 5 wells per dilution. The positive (anti-GAstV) egg yolk was prepared from GAstV-vaccinated chickens, and the negative was obtained from the pre-immunized and control groups of chickens. The mixture, 100 μL egg yolk dilution and the equal volume of GAstV (100 TCID_50_ in 50 µL), was incubated for 1 h at 37 °C before being added into monolayer LMH cells. The plates were incubated for 72 h in a 5% CO_2_ humid chamber at 37 °C. Cells were fixed by ice-cold Methanol/Acetone (Sinopharm Chemical Reagent Co., Ltd., Shanghai, China) for 30 min. After extensive washing, a mouse polyclonal anti-GAstV-Cap (prepared and preserved in our lab) was added to the wells and incubated at 4 °C overnight with 1:500 dilution. After extensive washing, a goat anti-mouse IgG (H+L) Alexa Fluor 488 (Beyotime, Shanghai, China) at a dilution of 1:500 was used as a secondary antibody. After washing, the images were acquired using a fluorescence microscope. The Reed–Muench method was used to calculate the virus neutralization titer for the samples [32,33].

### 2.8. Protection of Goslings Inoculated with GAstV

#### 2.8.1. Determination of Minimum Effective Dose

To determine the minimum effective dose of the anti-GAstV IgY in goslings, 55 one-day-old goslings were divided into five groups (groups 1–5 (G1–G5), *n* = 11 per group). Four experiment groups (G1–G5) were administrated with 0.3 mL (≈0.8 mg), 0.6 mL (≈1.6 mg), 0.9 mL (≈2.5 mg), and 1.2 mL (≈3.3 mg) of the anti-GAstV IgY subcutaneously, respectively. G5 was an untreated group. All goslings were challenged with 10^5.84^ TCID_50_ GAstV after 1 day post-immunization and monitored the clinical signs and mortality for 14 days. Cloacal swabs were collected at 4, 8, and 12 dpi. All goslings were euthanized at 14 dpi and the liver, spleen, and kidney samples were collected and fixed for subsequent staining with H&E.

#### 2.8.2. Estimation of Prophylactic Antiviral Treatment and Post-Infection Treatment

The prophylactic antiviral treatment (PAT) and post-infection treatment (PIT) were conducted to test the efficacy of anti-GAstV IgY. In the PAT experiment, 1-day-old goslings were administrated with 0.9 mL of the IgY before being infected with GAstV-AHQJ18 to calculate the prophylactic protection rates. Seven IgY immunization groups (*n* = 10, groups 6–12 (G6–G12)) were administrated with 0.9 mL IgY subcutaneously on day 1 and infected with 10^5.84^ TCID_50_ of the live GAstV-AHQJ18 after 6 h, 12 h, 1 d, 3 d, 5 d, 7 d, and 9 d, while the corresponding challenge control groups (*n* = 5, groups 13–19 (G13–G19)) were administrated with PBS on day 1 and challenged in the same manner.

In the PIT experiment, 25 1-day-old goslings were divided into five groups (*n* = 5, groups 20–24 (G20–G24)). All goslings were infected with 10^5.84^ TCID_50_ of the GAstV-AHQJ18 subcutaneously on day 1. Groups 20–23 were administrated with 0.9 mL of the IgY after 0, 1, 3, and 5 d. Group 24 (G24) served as a challenge control, only being infected with GAstV.

Every group of goslings was reared in isolation for 14 days after being infected with the GAstV-AHQJ18. Cloaca swabs were collected on 4, 8, and 12 dpi during the animal experiments to measure the protective rate using qRT-PCR.

### 2.9. qRT-PCR Analysis to Determine Viral Loads

Primers for qRT-PCR analysis of GAstV (forward: 5′-ACACCAGCGAGTATCTAGGC-3′, reverse: 5′-GTCGTATCCGCCAGAAGAGA-3′) were synthesized and qRT-PCR analysis was performed to detect the viral loads in cloaca swabs and organs samples as described previously using the Thermal Cycler Dice™ Real-Time System (Takara Bio Inc., Dalian, China [34].

### 2.10. Statistical Analysis

GraphPad Prism 7.0 was used for statistical analysis and figure drawing. All data are presented as the mean or the mean ± standard deviation (SD) of five goslings. One-way ANOVA and log-rank (Mantel-Cox) test were used to evaluate differences in the experimental data. Statistical significance set at *p* < 0.05.

## 3. Results

### 3.1. Phylogenetic and Pathogenicity Analysis of GAstV-AHQJ18

The GAstV-AHQJ18 strain belonged to the goose astrovirus type 2 (GAstV-2) according to the phylogenetic analyses, based on the whole genome sequences, ORF1a, ORF1b, ORF2 (Appendix A) and amino acids (Appendix A). The death of goslings occurred in the GAstV-infected group at 3 dpi and 6 dpi, one and two goslings appeared respectively (Figure 1A). Necropsy results showed that the typical gout pathological changes were found in the joints and viscera (Figure 1C). Histopathological examination showed necrosis and inflammatory cells in liver and spleen, and the damage to the brush border of renal tubules in the infected groups (Figure 1D). The detection of viral load from cloacal swabs indicated the shedding peak at around 8 dpi (Figure 1B). No mortality or any clinical signs were found in the control group.

### 3.2. Determination of IgY Titer against GAstV

The mixture of egg yolk was tested using AGID and VN assays. The fluctuation tendency of both techniques is consistent. The anti-GAstV IgY titer began to rise on day 7 after the primary immunization, peaked on day 49, and remained steady from day 49 to day 77. Eggs were collected from day 49 to day 77 (Figure 2A,B). SDS-PAGE was used to assess the purity and molecular weights of IgY antibodies against GAstV, revealing two bands of approximately 70 kDa and 27 kDa, respectively (Figure 2C). The purified IgY had a concentration of 2.79 mg/mL. The IFA results of the neutralization activity of anti-GAstV IgY demonstrated that the specific IgY was able to completely neutralize the GAstV at the VN titer of 2^13^ (17 ng) (Figure 2D). The neutralizing antibody titer of purified anti-GAstV IgY was 2^13.159^, calculated by the Reed–Muench method.

### 3.3. Protection of Goslings Inoculated with Anti-GAstV IgY

#### 3.3.1. Determination of Minimum Protective Dose of IgY

The survival rates of the G1–G5 were 90.91% (10/11), 81.8% (9/11), 100% (11/11), 100% (11/11), and 72.7% (8/11), respectively (Figure 3E). The viral load of cloacal swabs in G2–G4 significantly decreased at 8 dpi (*p* < 0.0001) compared to the control group, while G1 significantly increased (Figure 3A; *p* < 0.0001). In addition, the viral loads of G4 were significantly lower than that of G3 and G2 (Figure 3A). In G2–G4, the viral load in the liver, spleen, and kidney at 12 dpi were significantly lower than that of the control group (*p* < 0.0001), while G1 was higher than G5 (*p* < 0.0001; Figure 3B–D). Furthermore, there was no significant difference in liver viral load between G3 and G4 (*p* > 0.05; Figure 3B). Additionally, serious pathological damage to the liver and kidney was found in G1 and G5, including renal tubular epithelial cells degeneration and necrosis, destruction of the brush border structure of the proximal convoluted tubule, inflammatory cell infiltration in the renal interstitium, and necrosis and inflammatory in the liver (Figure 3F). The protection rates of G1–G4 were 54.5% (6/11), 72.7% (8/11), 90.9% (10/11), and 100% (11/11), respectively, whereas the infection rate in the control group was 63.6% (7/11) (Table 1). In general, 0.9 mL of the IgY (about 2.5 mg), with a protection rate of more than 80%, was designated as the minimum protective dose.

#### 3.3.2. Prophylactic and Curative Effect of IgY against GAstV

The protection rates were 90% (9/10), 90% (9/10), 100% (10/10), 100% (10/10), 80% (8/10), 60% (6/10), and 40% (4/10) in G6–G12, respectively (Table 2). The survival rates of the G6-G9 (IgY immunization groups) were higher than their corresponding control groups (Figure 4). The results suggested that more than 80% 1-day-old goslings were protected by administrated with 2.5 mg anti-GAstV IgY within 5 days.

The protection rates of G20–G23 were 100% (5/5), 80% (4/5), 40% (2/5), and 0 (0/5), respectively (Table 3). The survival rates of the G20–G23 were higher than the G24 (Figure 5). The results demonstrated that the administration of 2.5 mg IgY had a good curative effect. 

## 4. Discussion

In China, a goose-derived novel pathogen of gout has been identified as GAstV, which mainly infected 1- to 20-day-old goslings [34]. Goslings are more susceptible to GAstV infection than older geese, and 25- to 35-day infected geese exhibited mild symptoms [34]. GAstV can be transmitted horizontally and vertically, and has the possibility of cross-species transmission, causing significant economic losses to the goose industry [35]. In recent years, multiple investigations have been conducted, mainly focusing on the development of methods for molecular biology and immunology to identify the GAstV [36,37,38]. The application of traditional chemotherapy was unavailable, which may further increase the burden of kidney metabolism and aggravate the course of the disease as the GAstV infection caused severe damage to the structure and function of the kidney [39]. In addition, there are no commercial vaccines has been approved so far.

In recent years, specific yolk IgY antibodies have been widely used in the prevention and treatment of infectious diseases in animals, mainly because IgY antibodies are easy to prepare, safe, inexpensive, and have no known side effects [40]. Passive immunization includes maternal antibodies and artificial passive immunization, which neutralizes the virus by injecting specific antibodies to prevent virus infection [41]. The prophylactic and therapeutic effect of IgY against viral infections in poultry has been extensively investigated. The maternal antibodies can prevent the disease for the first 2–3 weeks of age in avian pathogenic virus, including NDV, IBDV, influenza, gosling parvovirus (GPV), duck astrovirus (DHV), and reovirus. IgY against GPV, the causative agent of the gosling plague (GP), was able to prevent GP development in 1-day-old and 3-day-old goslings to 5-day-old and 7-day-old, respectively [42]. Intramuscular administration of the anti- DHAV IgY was performed on ducklings infected by the virus. The results showed that the cure rate of duck hepatitis 1 was 100% after 12 and 18 h of artificial infection [43]. In the present study, the results of the prevention and the treatment experiments showed the prevention and cure rates are over 80% when yolk antibody was performed in the early stages of the GAstV infection. These results illustrated that the specific IgY can be a new strategy for protecting goslings from GAstV infection.

Antigen type is one of the most important factors in the development and production of specific IgY [44]. The current study used the GAstV-AHQJ18 strain to prepare the inactivated antigen. The results of phylogenetic and pathogenicity analysis of GAstV-AHQJ18 strain showed that it is a virulent strain of the genotype GAstV-2 strain, which are more widespread genotype in China, suggesting that GAstV-2 strain has a wide application value for preparing the specific IgY. In this study, the inactivated virus produced a high AGID titer and VN titer of the IgY, which significantly protects goslings from GAstV infection.

Many studies of the Fc receptor for immunoglobulins in cattle, sheep, pigs, and horses have been cloned and characterized recently [45]. These findings showed the Fc may have specific diversity in binding sites [45]. The Fc regions of the IgY are similar to the crystal structure of free IgE-Fc, and both Fc are N-glycosylated in their CH3 domains with high-mannose-type oligosaccharide [46,47]. Multiple studies indicated that the astrovirus could activate the immune evasion mechanism or immune supersession by complement or cytokines to enhance viral replication. The human astrovirus (HAstV) coat protein binds C1q and mannose-binding lectin (MBL) resulting in the inhibition of classical and lectin pathways of complement, respectively [48]. Studies in goose and turkey poults suggest that astrovirus infection suppresses the immune system, making the host susceptible to subsequent infections [49]. Wu et al. found the increasing level of TGF-β in spleen and kidney indicates that GAstV infection may induce immune suppression [50]. In this study, the specific IgY showed a high VN titer in vitro and can fully neutralize GAstV, while in vivo, the protective effect was not particularly significant. We suspect that the complement and cytokines involved the immune evasion mechanism and immune supersession in vivo GAstV infection. In addition, the condition of the models in vitro studies is single and controllable, while the in vivo animal studies are complex and variable.

It is critical to employ the IgY to prevent early infection since goslings are the major susceptible animals. One-day-old goslings were used as research subjects in this study. Both prevention and treatment could offer effective protection. However, the results showed the groups (G1 and G2) inoculated with less IgY led to a decline in the protection rate or aggravated the viral load, compared to the control group (G5), which might be antibody-dependent enhancement (ADE). However, no study has confirmed this conjecture, highlighting the need for further studies to be carried out in this area.

## 5. Conclusions

This study proposed a strategy for preventing and treating the GAstV infection. The anti-GAstV IgY was produced by the immunization of laying hens with the inactivated GAstV, and it could neutralize the GAstV-AHQJ18 in vivo and had a good prophylactic and therapeutic protection for the early GAstV infection.

## Figures and Tables

**Figure 1 vaccines-10-02090-f001:**
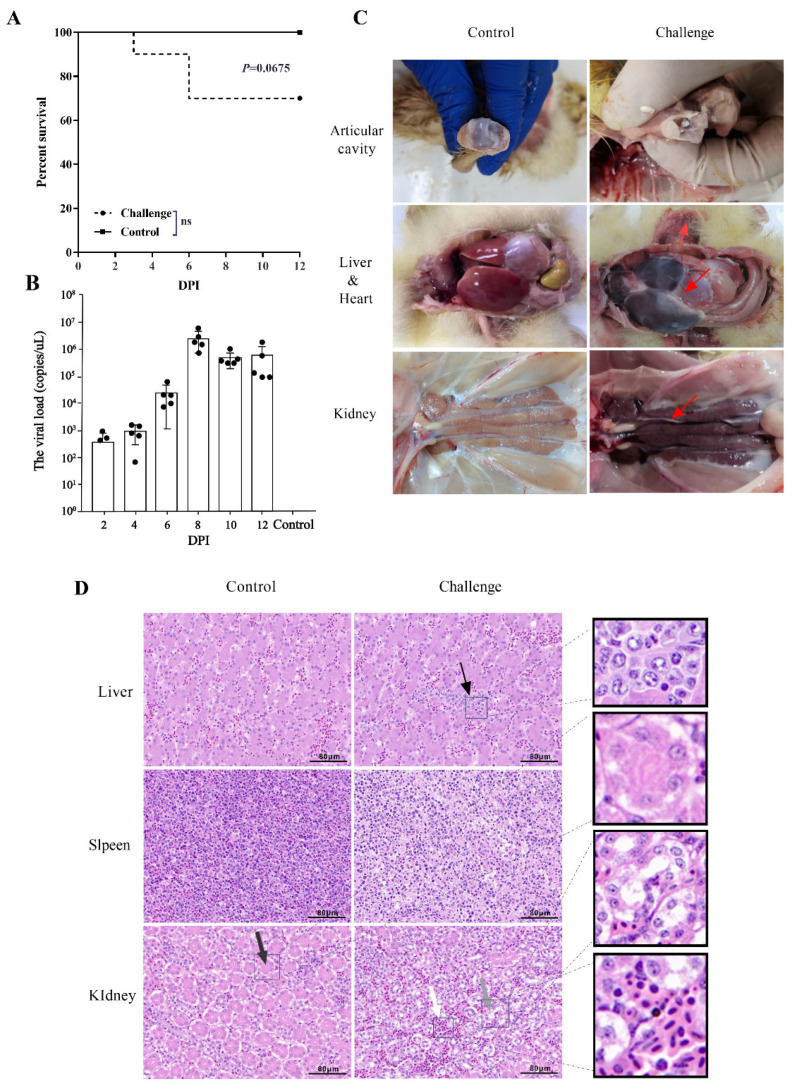
Pathogenicity analysis of GAstV-AHQJ18 strain. (**A**) The survival rates were monitored daily. Survival data were analyzed by log-rank (Mantel–Cox) test. *p* values: no significant difference (ns) *p* > 0.05. (**B**) Viral load in cloacal swabs were detected at 2, 4, 6, 8, 10, and 12 dpi. (**C**) Postmortem lesions of goslings infected GAstV-AHQJ18 strain. There were hepatonephromegaly and urate deposits in the articular cavity, peritoneum, crureus, and kidney. Red arrow: urate deposits. (**D**) Histopathological changes in liver, spleen and kidney in goslings. The histopathological changes in the control group were normal. Liver and kidney damage were observed in the infected group. Thin black arrow: necrosis and inflammatory infiltration in liver. Thick black arrow: the normal brush border of the proximal tubule in the control group. Gray arrow: the villi structure disappeared in the infected group. White arrow: lymphocyte infiltration. Scale bar = 80 μm.

**Figure 2 vaccines-10-02090-f002:**
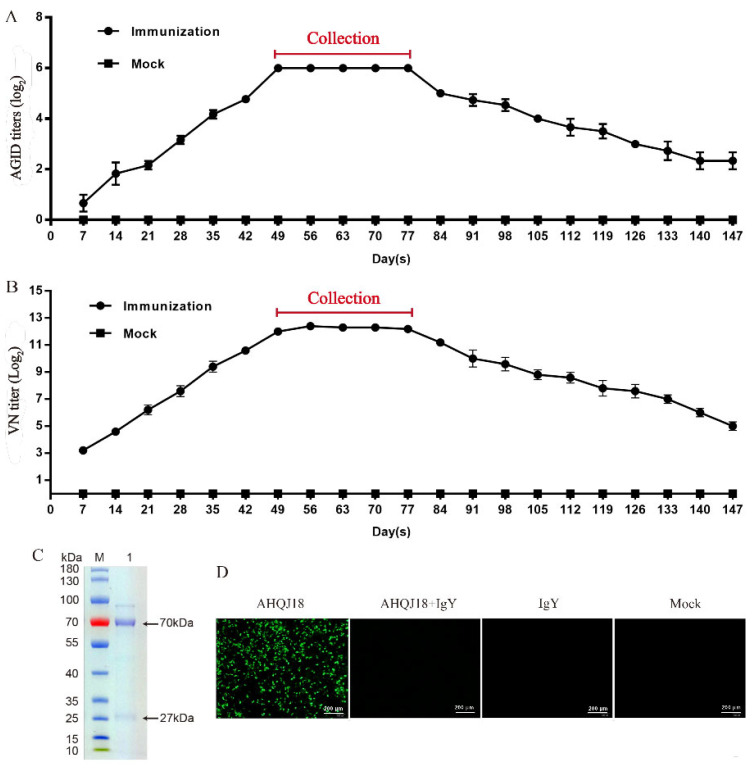
Preparation of anti-GAstV IgY. Antibody dynamics and collection period were determined by AGID (**A**) and VN (**B**). Red line: the collection period of the IgY. (**C**) The molecular weights and purity of the anti-GAstV IgY were identified by SDS-PAGE. M: Protein molecular marker. 1: IgY extracted by PEG 6000. Black arrow: the molecular weights of the heavy chains and light chains. (**D**) The fluorescence signals of the group of GAstV-AHQJ18, 2^13^ dilutions of IgY incubated with GAstV-AHQJ18 strain, IgY, and mock were incubated in 37 °C incubator for 3 days. The fluorescence signals of GAstV-AHQJ18 were detected by fluorescence microscope (×200). Scale bar = 200 μm.

**Figure 3 vaccines-10-02090-f003:**
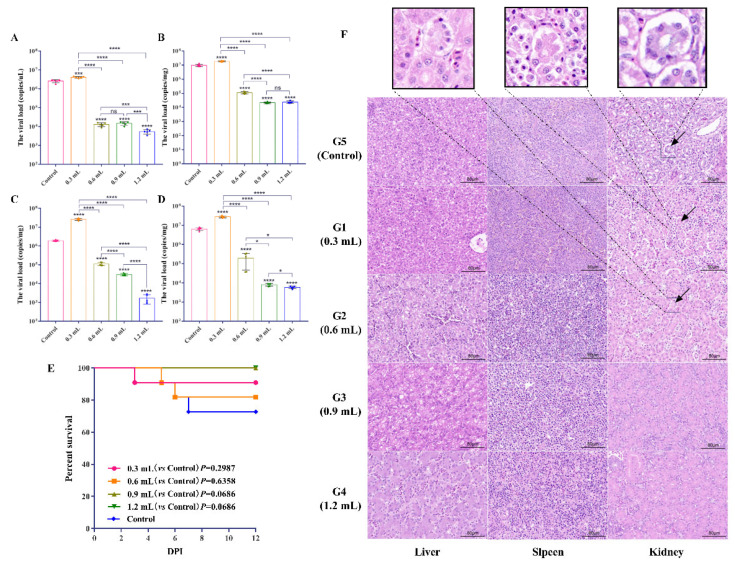
Estimation of the preventive effect of different doses of the anti-GAstV IgY. The mRNA levels of GAstV-AHQJ18 in cloacal swabs (**A**), liver (**B**), spleen (**C**), and kidney (**D**). The survival rates of G1–G5 were monitored daily for 12 days (**E**). Survival data were analyzed by log-rank (Mantel–Cox) test. Pathological changes in the infected goslings in G1–G5 (**F**). Necrosis and inflammatory infiltration in liver (H&E) were observed in G1, G2, and G5. Kidney (H&E) in groups G1, G2, and G5, renal tubular epithelial cells severe degeneration and necrosis, and the brush border structure of the proximal convoluted tubule was destroyed (black arrow). Kidney (H&E) in groups G3 and G4, brush border structure of the proximal convoluted tubule intact. In all panels with *p* values are expressed as mean ± SD, *n* = 5. * *p* < 0.05; *** *p* < 0.001, **** *p* < 0.0001, ns *p* > 0.05. Scale bar = 80 μm.

**Figure 4 vaccines-10-02090-f004:**
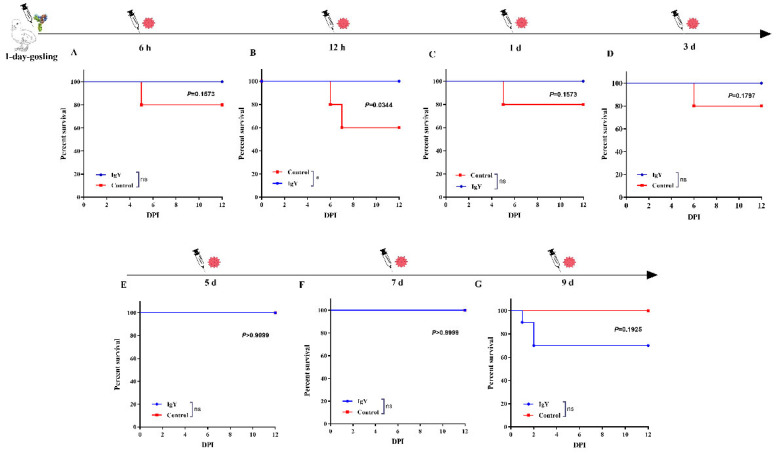
The survival rates of the G6–G19 (**A**–**G**). G6–G12 were subcutaneously inoculated with 0.9 mL IgY on day 1, while G13–G19 were corresponding control groups administrated with an equal volume of PBS, and subsequently, each group was challenged with 10^5.84^ TCID_50_ of the live GAstV at 6 h, 12 h, 1 d, 3 d, 5 d, 7 d, and 9 d post inoculation subcutaneously. The survival rates of G6–G19 were monitored daily. Survival data were analyzed by log-rank (Mantel–Cox) test. *p* values are expressed as mean ± SD, * *p* < 0.05; ns *p* > 0.05.

**Figure 5 vaccines-10-02090-f005:**
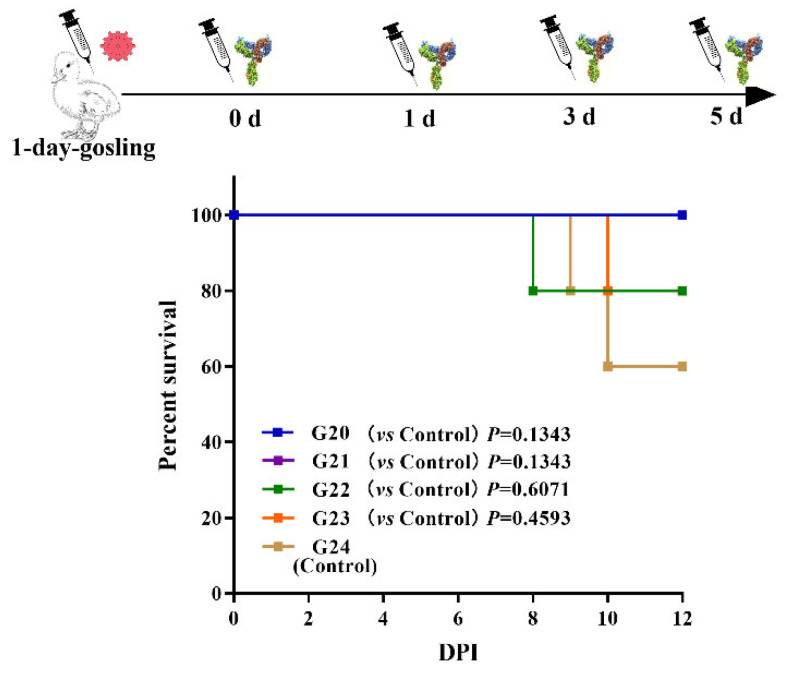
The survival rates of the G20–G24. G20–G24 were infected with GAstV at 10^5.84^ TCID_50_ subcutaneously on day 1. G20–G23 were inoculated with the anti-GAstV IgY at 0, 1, 3, and 5 dpi. G24 was a control group. The survival rates of G20–G24 were monitored daily. Data were analyzed by log-rank (Mantel–Cox) test. *p* values are expressed as mean ± SD, *n* = 5. ns *p* > 0.05.

**Table 1 vaccines-10-02090-t001:** Estimation of the minimum protective dose of yolk antibody.

Group	Dose (mL)	Number of Samples	Number of Infections	Ratio of Positive to Total Samples (a/c)	Ratio of Negative to Total Samples (b/c)
1	0.3	11	5	5/11	6/11
2	0.6	11	3	3/11	8/11
3	0.9	11	1	1/11	10/11
4	1.2	11	0	0/11	11/11
5	PBS	11	7	7/11	4/11

a, b, and c represent the number of infection, healthy, and total goslings, respectively.

**Table 2 vaccines-10-02090-t002:** Protection rates of the prophylactic protection test with the anti-GAstV IgY.

Group	Number of Samples	The Ratio of Positive to Negative Samples at Different Time Points (a/b)			Ratio of Negative to Total Samples (b/c)
		4 d	8 d	12 d	
6	10	0/10	1/9	1/9	9/10
13	5	2/3	1/3	1/3	3/5
7	10	0/10	1/9	1/9	9/10
14	5	4/1	3/0	3/0	0/5
8	10	0/10	0/10	0/10	10/10
15	5	3/2	2/2	2/2	2/5
9	10	0/10	0/10	0/10	10/10
16	5	2/3	2/2	4/0	0/5
10	10	1/9	2/8	2/8	8/10
17	5	2/3	3/2	5/0	0/5
11	10	3/7	4/6	4/6	6/10
18	5	2/3	3/2	3/2	2/5
12	10	1/6	3/4	3/4	4/10
19	5	2/3	3/2	3/2	2/5

a, b, and c represent the number of infection, healthy, and total goslings, respectively.

**Table 3 vaccines-10-02090-t003:** Protection rates of the therapeutic protection test with anti-GAstV IgY.

Group	Number of Samples	Number of Infections	The Ratio of Positive to Negative Samples at Different Time Points (a/b)			Ratio of Negative to TOTAL Samples (b/c)
			4 d	8 d	12 d	
20	5	0	0/5	0/5	0/5	5/5
21	5	1	1/4	1/4	1/4	4/5
22	5	3	1/4	3/2	3/2	2/5
23	5	5	1/4	5/0	5/0	0/5
24	5	5	5/0	5/0	5/0	0/5

a, b, and c represent the number of infection, healthy, and total goslings, respectively.

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
