# Peer review of "An IgY Effectively Prevents Goslings from Virulent GAstV Infection"

_vaccines, 2022, doi:10.3390/vaccines10122090_

Round 1

Reviewer 1 Report

In the manuscript entitled” An IgY Effectively Prevent Goslings from Virulent GAstV in- 2 fection”, Mengran Zhang and colleagues prepared hens-derived anti-GAstV IgY, and found it showed good prophylactic and therapeutic effects against GAstV infection, which provide a new strategy for preventing and treating GAstV infection. The topic is of certain interest, while the manuscript was not very presented, a number of issues have to be addressed before publication.

1.       Figure1F, why the control has a virus titer? Please check.

2.       Figure2, the VN titer was much higher than AGID titer, is it normal or not? In Figure2D, which concentration of the specific IgY was used? Since the IgY totally neutralized GAstV and no fluorescence signal appeared.

3.       In vitro assays, the specific IgY showed a high VN titer and can fully neutralized GAstV, while in vivo, the protective effect was not particularly significant, why? This should be discussed.

4.       Figure1A-D, the picture is too compressed to see clearly.

5.       Higher magnification images or partial enlarged view should be provided to illustrate the lesions indicated by the arrows in figure1H and 3H.

6.       Figure1E and figure3E, both are survivorship curves, no clinical signs were presented, so the figure legends are not suitable.

7.       Figure4 and figure5, the legends are too simple.

8.       All the survivorship curves lack statistical analysis.

9.       Please check the references format carefully, for example, reference 24 and 26.

Author Response

Thanks for your letter and for the reviewers’ comments concerning our manuscript entitled “An IgY Effectively Prevent Goslings from Virulent GAstV infection” (Manuscript ID: vaccines-2045265). Those comments are all valuable and helpful for revising and improving our paper. We have studied all comments carefully and have made conscientious corrections. Revised portions are marked in the paper. The main corrections in the paper and the responses to the reviewers’ comments are as follows.

Response to reviewers 1:

In the manuscript entitled” An IgY Effectively Prevent Goslings from Virulent GAstV infections”, Mengran Zhang and colleagues prepared hens-derived anti-GAstV IgY, and found it showed good prophylactic and therapeutic effects against GAstV infection, which provide a new strategy for preventing and treating GAstV infection. The topic is of certain interest, while the manuscript was not very presented, a number of issues have to be addressed before publication. Overall, this manuscript is well described. However, several concerning points need to be addressed.

Response:

Thank you very much for your comments. The red part has been revised according to your comments. Revision notes, point-to-point, are listed as follows.

  1. Figure1F, why the control has a virus titer? Please check.

Response:

Thanks for your question. The SYBR Green â…  real-time PCR method, established in the reference, showed the number of cycles were 35[1]. However, the cycles of the instrument were set to 40, and the control group had a crest in 36-38 cycles, which were non-specific crest. No viral RNA was detected in the control group within 35 cycles, and we redraw the Figure IF, which renamed Figure 1B in the manuscript.

[1] Luo, Y., Isolation and identification of goose astrovirus and establishment of Real-time quantitative PCR detection method. MA thesis, Nanjing Agricultural University, Nanjing, China, 2020.

  1. Figure2, the VN titer was much higher than AGID titer, is it normal or not? In Figure2D, which concentration of the specific IgY was used? Since the IgY totally neutralized GAstV and no fluorescence signal appeared.

Response:

Thanks for raising these questions. The VN titer was higher than the AGID titer is normal. In the AGID, 70 uL of the 100-fold concentrated GAstV (7×106.84 TCID50) was deposited in the center well, while 200 TCID50 GAstV was used in the VN assay. The more antigens, the more concentration of the IgY is needed, and the IgY dilution in the AGID is much lower than in the VN.

Each well used 17 ng of the specific IgY, representing a concentration of 2.79 mg/mL specific IgY was diluted by 1013 in Figure 2D.

  1. In vitro assays, the specific IgY showed a high VN titer and can fully neutralized GAstV, while in vivo, the protective effect was not particularly significant, why? This should be discussed.

Response: Thanks for your comment, it is a worth discussing question. Many studies of the Fc receptor for immunoglobulins in cattle, sheep, pigs and horses have been cloned and characterized recently[42]. These findings showed the Fc may have specific diversity in binding sites[42]. The Fc regions of the IgY are similar to the crystal structure of free IgE-Fc, and both Fc are N-glycosylated in their CH3 domains with high-mannose-type oligosaccharide[43, 44]. Multiple studies indicated that the astrovirus could activate the immune evasion mechanism or immune supersession by complement or cytokines to enhance viral replication. The human astrovirus (HAstV) coat protein binds C1q and mannose-binding lectin (MBL) resulting in the inhibition of classical and lectin pathways of complement, respectively[45]. Studies in goose and turkey poults suggest that astrovirus infection suppresses the immune system, making the host susceptible to sequent infections[46]. Wu et al. found the increasing level of TGF-β in spleen and kidney indicates that GAstV infection may induce immune suppression[47]. In this study, the specific IgY showed a high VN titer in vitro and can fully neutralize GAstV, while in vivo, the protective effect was not particularly significant. We suspect that the complement and cytokines involved the immune evasion mechanism and immune supersession in vivo GAstV infection. In addition, the condition of the models in vitro studies is single and controllable, while the in vivo animal studies are complex and variable.

In addition, we have added the part to the manuscript.

  1. Figure1A-D, the picture is too compressed to see clearly.

Response: Thanks for the suggestion. We have regrouped Figure 1A-D in supplementary document, and renamed Figure S1.

  1. Higher magnification images or partial enlarged view should be provided to illustrate the lesions indicated by the arrows in figure1H and 3H.

Response: Thanks for the suggestion, it was corrected as suggested.

  1. Figure1E and figure3E, both are survivorship curves, no clinical signs were presented, so the figure legends are not suitable.

Response: Thanks for the reminder, it was corrected as suggested.

  1. Figure4 and figure5, the legends are too simple.

Response: Thanks for the reminder, it was corrected as suggested.

  1. All the survivorship curves lack statistical analysis.

Response: Thanks for the reminder, it was corrected as suggested.

  1. Please check the references format carefully, for example, reference 24 and 26.

Response: Thanks for the reminder, it was corrected as suggested.

Reviewer 2 Report

Goose astrovirus (GAstV) leads to high mortality in 1- to 20-day old goslings, which posing a severe threat to entire geese groups. However, there are still no available vaccines for use. So, it is very important to develop novel methods to control this viral infections. IgY is an effective measure to control different diseases. In this study, an effective IgY was developed to control goose astrovirus, which will lay foundation for further control this disease.

Major revisions

1. English is not very fluent, it should be polished by native speaker.

2. The method of immunization was not indicated in the abstract.

Minor revisions

Line 17, PEG-600 precipitation.

Line 19 of should be replaced with are.

Line 23, goose should be Goose.

Line 42, widely used for what?

Line 67, in a LMH cells and titers were determined as……

Line 76, reference is lacking.

Line 81, isolated should be replace with separated.

Line 90, viral aggregates?

Line 120, references for 25 and 26 should not be separated.

Line 163, slaughtered humanely should be euthanized.

Line 170-171, the meaning of this sentence is not clear.

Line 174-178, the groups are not mached.

Line 195, delete “was”.

Line 198, the indication of (1) and (2) is not clear.

Line 208, the version of MEGA in text is not united.

Line 221, initial should be replaced with primary.

Reviewer 3 Report

The manuscript “An IgY effectively prevents goslings from Virulent GAstV infection”, has tried to address the problem related to early mortality and morbidity in gooselings in the Geese industry due to GAstV.

Authors have used purified IgY from yolk of the immunized eggs and used it to immunize the bird and did challenge study.

I have the suggestions below to improve the manuscripts.

1. In the introduction section provide some background about the vaccine used to control GAstV in China or elsewhere and why this strategy is important to control this disease.

2. What gap this IgY prophylactic strategy will fill is not previously addressed by other disease mitigation strategies in GAstV.

3. Line 48 newcastle should have “N” uppercase

4. In Materials and methods section, sub section 2.2 dentification and Pathogenicity of AHQJ18 Strain 

is too descriptive.In previous studies, if  it had been WG sequenced and pathogenicity study had been made, then this section is redundant.

5. Line 77 has no reference for Xios pathogenicity study.

6. Section 2.4 of materials and methods, any reference for selecting 3 sites with different doses of antigen for immunization of hens.

7. Section 2.5 of materials and methods line 113 Sykes (Reference ) is vague and no specific chapter or page has been mentioned for AGID assay.

8. Section 2.8 of materials and methods line 135 references 27 & 28 not specifically related to VN methods for IgY rather it looks like general VN references. Please clarify it.

9. Line 144-146, why VN titer is not calculated as complete inhibition in all wells of highest dilution, rather only half the wells, any specific reference related to it.

10. Line 152-153 has no reference to the Reed and Munch method and this method was used to calculate TCID50 or to calculate titer.

11. Sections 2.8.1 and 2.8.2 are confusing to the reader and provide a tabular or graphic representation of groups.

12. Line 171, 7d is repeated twice

13. Line 176 before group 20-24 is mentioned in PIT study and again, group 30-33 is mentioned, which has not been defined before. Please clarify this.  Is this a typo and do you want to say group 20-23?

14.In result section 3.1 phylogenetic tree data can be put in supplementary table 1.

15. Table 1 is a suppl table, so mention it clearly in line 197.

16. Line 226-227, which’s the highest dilution, IgY completely inactivated the GAstV.

17. Line 233 Figure 2D, provides magnification and scale.

18. Line 237 typo strain

19. The discussion section is very brief and the result is not properly discussed.

Round 2

Reviewer 3 Report

Overall authors have addressed the reviewers concern , some minor spelling mistakes need to corrected

Line 308_ tragedy to strategy

Line 327 _ sequent to subsequent